# Characterization of Genetically Modified Microorganisms Using Short- and Long-Read Whole-Genome Sequencing Reveals Contaminations of Related Origin in Multiple Commercial Food Enzyme Products

**DOI:** 10.3390/foods10112637

**Published:** 2021-10-30

**Authors:** Jolien D’aes, Marie-Alice Fraiture, Bert Bogaerts, Sigrid C. J. De Keersmaecker, Nancy H. C. Roosens, Kevin Vanneste

**Affiliations:** 1Transversal Activities in Applied Genomics (TAG), Department Expertise and Service Provision, Sciensano, J. Wytsmanstraat 14, 1050 Brussels, Belgium; Jolien.Daes@sciensano.be (J.D.); Marie-Alice.Fraiture@sciensano.be (M.-A.F.); Bert.Bogaerts@sciensano.be (B.B.); Sigrid.DeKeersmaecker@sciensano.be (S.C.J.D.K.); Nancy.Roosens@sciensano.be (N.H.C.R.); 2Department of Plant Biotechnology and Bioinformatics, Ghent University, 9000 Ghent, Belgium

**Keywords:** genetically modified microorganism (GMM), *Bacillus velezensis*, food enzyme, whole-genome sequencing, hybrid genome assembly, SNP phylogenomic analysis

## Abstract

Despite their presence being unauthorized on the European market, contaminations with genetically modified (GM) microorganisms have repeatedly been reported in diverse commercial microbial fermentation produce types. Several of these contaminations are related to a GM *Bacillus velezensis* used to synthesize a food enzyme protease, for which genomic characterization remains currently incomplete, and it is unknown whether these contaminations have a common origin. In this study, GM *B. velezensis* isolates from multiple food enzyme products were characterized by short- and long-read whole-genome sequencing (WGS), demonstrating that they harbor a free recombinant pUB110-derived plasmid carrying antimicrobial resistance genes. Additionally, single-nucleotide polymorphism (SNP) and whole-genome based comparative analyses showed that the isolates likely originate from the same parental GM strain. This study highlights the added value of a hybrid WGS approach for accurate genomic characterization of GMM (e.g., genomic location of the transgenic construct), and of SNP-based phylogenomic analysis for source-tracking of GMM.

## 1. Introduction

Enzymes, additives and flavorings produced by microbial fermentation are widely used and indispensable for the food and feed industry. Genetically modified microorganisms (GMM) are frequently employed to increase microbial enzyme production efficiency and/or yield [1]. However, their presence is unauthorized in the final products commercialized in the European Union (EU) food and feed chain (EC/2003/1830). Moreover, since these GMM usually carry antimicrobial resistance (AMR) genes used as selection markers, the ingestion of such contaminated products has raised health concerns related to potential AMR horizontal gene transfer to pathogens and other gut microbiota.

Several unexpected contaminations of GMM carrying AMR genes, consisting of DNA as well as living GMM, were recently reported in microbial fermentation food and feed products commercialized on the EU market [2,3,4,5,6,7]. Control of potential GMM contaminations in microbial fermentation products is therefore crucial. In contrast to genetically modified organisms (GMO) directly intended for human and animal consumption, manufacturers of GMM-derived fermentation products do not need to provide GMO detection methods. To ensure food safety and traceability, as well as the consumer’s freedom of choice, we have recently developed a GMM detection strategy using real-time PCR, a technology commonly mastered by enforcement laboratories performing GMO routine analysis. GMM presence is first screened for by targeting sequences frequently found in GMM, including three AMR genes (*cat*, *aadD* and *tet-l*, encoding chloramphenicol, kanamycin and tetracyclin resistance, respectively) and the pUB110 shuttle vector carrying *aadD*. Following a positive signal for at least one of these screening markers, the presence of specific GMM is investigated by targeting their unnatural associations [2,3,4,5,6,7].

Although such a qPCR strategy allows to prove the presence of a (specific) GMM in a given sample, further investigation is needed to obtain accurate risk assessment. More specifically, if a transgenic construct carrying AMR genes is harbored in the host on a free plasmid, the risk for horizontal transfer of AMR to other, potentially harmful, microorganisms may be greater than when the same transgenic construct is integrated into the microbial host genome. To perform in-depth genetic characterization, short-read whole-genome sequencing (WGS) has previously been used on a vitamin B2-producing GM *Bacillus subtilis* strain (RASFF2014.1249) [7,8], and the protease-producing GM *B. velezensis* that is the focus of this study (RASFF2019.3332) [2]. Moreover, the benefit of additionally using long-read WGS data in a hybrid assembly approach was successfully demonstrated for the vitamin B2-producing GM *B. subtilis* strain [9]. However, despite these successful case studies, the use of WGS for the characterization of GMM is currently still very limited.

Furthermore, no strategy is currently available to determine potential links between different contaminated products. Bacteria are genetically more flexible than plant or animal genomes. This can allow, for example, transformation of multiple parental strains with the same recombinant plasmid. Targeting an unnatural association on a free plasmid does not allow to differentiate between potentially different host strains, even from different species. Additionally, due to the variable possible outcomes of the recombination process leading to genomic integration of a construct, targeting the insertion of the construct in the genome does not provide a unique signature for a specific GMM, contrary to using an event-specific qPCR assay covering the unnatural association for GM plants. For source-tracking of bacterial outbreaks, e.g., identification of contaminated food leading to a food-borne outbreak, the current state-of-the-art approach is WGS of potential bacterial causal agents that were isolated from both the patients and the material, e.g., foodstuff, under suspicion, followed by phylogenetic analysis based on single-nucleotide polymorphisms (SNPs) [10,11]. A similar strategy might be used to investigate whether GMM contaminations originate from a common ‘source’, which in this case would pertain to a common parental GMM host strain used to manufacture different fermentation products.

Using WGS, this study reports an in-depth genomic characterization and phylogenomic comparison of 10 protease producing GM *B. velezensis* strains, carrying a transgenic construct derived from shuttle vector pUB110, that were isolated from four different commercial FE products. To unambiguously determine both the location and copy number of the transgenic construct, GM bacterial isolates were subjected to both short-read Illumina and long-read Oxford Nanopore Technology (ONT) sequencing to employ a *de novo* hybrid assembly strategy and additional in-depth bioinformatics investigation. We demonstrate that the GMM primarily carry the transgenic construct, with a single copy of the wild-type derived protease encoding gene, on a free high-copy pUB110-derived plasmid. Additionally, transient unstable integration of this transgenic construct into the chromosome can occur. Comparative phylogenomics analysis was performed to investigate the relationship between the GMM isolated from the different FE products, demonstrating that the obtained GMM were genetically almost identical, indicating that they likely originate from the same GMM strain and presumably manufacturer.

## 2. Materials and Methods

### 2.1. GMM Isolation from FE Products

Four FE products in solid form were collected from the market (Table 1). One gram of each batch tested from each FE product was added to 250 mL of Brain–Heart Infusion broth (Sigma-Aldrich) in the presence of Kanamycin (50 µg/mL; Sigma-Aldrich) for incubation overnight at 30 °C. An amount of 100 μL of the culture was plated on nutrient agar (Sigma-Aldrich) in the presence of Kanamycin (50 µg/mL; Sigma-Aldrich) for incubation overnight at 30 °C. Based on real-time PCR analysis (see Section 2.2), isolates of the GM *B. velezensis* producing protease were selected for genomic analysis. For each batch tested from each FE product, two isolates were selected, to investigate the presence of potentially different strains, as well as to serve as a back-up against potential loss of plasmids from the isolates.

### 2.2. Real-Time PCR Colony Assays

The selection of the bacterial isolates was determined using a real-time PCR marker (GMMprotease-R) specific to a GM *B. velezensis* producing protease developed and published previously (RASFF2019.3332) [2]. Each real-time PCR colony assay was performed in a standard 25 µL reaction volume containing 1X TaqMan^®^ PCR Mastermix (Diagenode, Liège, Belgium), 400 nM of each primer (Eurogentec, Liège, Belgium) and 200 nM of the probe (Eurogentec, Liège, Belgium). The real-time PCR program consisted of a single cycle of DNA polymerase activation for 10 min at 95 °C followed by 45 amplification cycles of 15 s at 95 °C (denaturing step) and 1 min at 60 °C (annealing-extension step). All runs were performed on a CFX96 Touch Real-Time PCR Detection System (BioRad, Hercules, CA, USA). For each assay, a positive control (genomic DNA from the protease producing GM *B. velezensis* reported by [2] and an NTC (no template control) was included.

### 2.3. DNA Extraction, Library Preparation and Whole Genome Sequencing

DNA from GMM bacterial isolates was extracted using Quick-DNA™ HMW MagBead Kit (Zymo Research, Irvine, CA, USA) according to the manufacturer’s instructions and then visualized by capillary electrophoresis using the Tapestation 4200 device with the associated genomic DNA Screen Tape and reagents (Agilent, Santa Clara, CA, USA) (Appendix A). Each DNA concentration was measured by spectrophotometry using the Nanodrop^®^ 2000 (ThermoFisher, Dover, DE, USA) and each DNA purity was evaluated using the A260/A280 and A260/A230 ratios.

Plasmid DNA from the GM bacterial isolate Cob9-1 (Table 1) was extracted using the QIAprep Spin Miniprep Kit (QIAGEN, Hilden, Germany) according to manufacturer’s instructions. For the plasmid DNA, only short-read sequencing was performed.

Short-read DNA libraries were prepared using the Nextera XT DNA library preparation kit (Illumina, San Diego, CA, USA) according to manufacturer’s instructions. Sequencing was carried out on an Illumina MiSeq system with the V3 chemistry, obtaining 250 bp paired-end reads, and aiming for a theoretical coverage of 60× per sample, based on the average *Bacillus* genome size of ~4 Mbp.

Long-read DNA libraries were prepared using the ligation sequencing kit (SQK-LSK109; Oxford Nanopore Technologies, Oxford, UK) and barcoding kit (EXP-NBD104; Oxford Nanopore Technologies, Oxford, UK) according to the manufacturer’s instructions. Two groups of five DNA libraries individually barcoded were prepared and each group was loaded on an individual R9 MinION flow cell to be sequenced for 48 h.

### 2.4. Genome Assembly and Characterization

Raw short reads were preprocessed with Trimmomatic 0.38 [12] with the following settings: ILLUMINACLIP:NexteraPE-PE.fa:2:30:10, LEADING:10, TRAILING:10, SLIDINGWINDOW:4:20, MINLEN:50. Quality of raw and preprocessed data was evaluated using FastQC 0.11.5 (www.bioinformatics.babraham.ac.uk/projects/fastqc, accessed on 14 August 2021)) with default settings.

Raw long reads were basecalled and demultiplexed with Guppy 4.2.3 in GPU mode. Guppy_barcoder was run with the—trim_barcodes option to remove barcodes. Filtlong 0.2.0 (github.com/rrwick/Filtlong, accessed on 15 August 2021) was applied to raw fastq data to remove reads with an average quality score below 7 and read lengths below 1000 bp. Quality statistics on raw and filtered data were collected with NanoPlot 1.33.0 [13] with default settings.

For long read assembly, Canu 1.8 [14] was used, with the genome size set at 4.5 Mbp and default settings. The resulting assembly was subjected to iterative short-read polishing with the Unicycler-polish module of Unicycler 0.4.7 [15], using default settings and the following dependencies: ALE v20180904 [16], Pilon 1.23 [17], Bowtie2 2.3.4.3 [18] and samtools 1.9 [19].

Hybrid assembly was carried out with Unicycler 0.4.7 and hybridSPAdes 3.13.0 [20]. The following Unicycler dependencies were employed: SPAdes, Pilon 1.23, Bowtie2 2.3.4.3, Racon 1.3.1 [21] and samtools 1.9. The Canu assembly was provided to Unicycler to be used instead of the default long read assembly that Unicycler produces internally, since we observed the final assembly being less fragmented using this approach. Unicycler uses this long read assembly in a hybrid strategy to improve the final assembly, which is based on both short and long reads. Otherwise, default settings were used. HybridSPAdes was run with the ‘--careful’ argument, k set to ‘21, 33, 55, 77, 99, 127′, and coverage cutoff set to 10.

The MiSeq data generated from the plasmid DNA extract was assembled with Unicycler in short-read mode, with default settings.

For subsequent whole genome comparison, a representative assembly was selected for each isolate, i.e., the Unicycler assembly when available, and the Canu assembly otherwise (Appendix A). 

Assembly statistics were obtained with Quast 5.0.2 [22], and assembly graph visualization with Bandage 0.8.1 [23]. Genome annotation was done with Prokka 1.11 [24], with default settings. Characterization of prophage sequences was conducted through the web interface of PHASTER [25]. Genotypic AMR detection was performed as described in Bogaerts et al. [26], with one modification, i.e., the National Database of Antibiotic Resistant Organisms (NDARO) (retrieved on 12 January 2021) was used instead of the ResFinder database.

### 2.5. Follow-Up Analysis of Long Reads

The workflow is represented as a flowchart in Appendix A. BLAST 2.7.1+ was run on raw long read datasets using pUB110 (Genbank: M19465.1) as query, increasing the maximum target sequences to 1 million to retain all possible hits. The resulting hits were filtered from the raw data with the filterbyname.sh script of bbtools 38.34 (Sourceforge.Net/Projects/Bbmap, accessed on 14 August 2021), with default settings, except for the addition of ‘include = t’ to retain hits matching the names instead of excluding them. Statistics on the filtered dataset were collected with NanoPlot.

A subset of obtained BLAST hits, with read length of at least 6756 bp, to exclude sequences that are smaller than the pUB110-derived recombinant plasmid length, was visualized with the web interface of Kablammo [27]. Specific reads were manually selected for further analysis with web-based BLAST against the NCBI nucleotide collection database. This analysis yielded a read of ~44 kbp in length originating from sample Crystal-1, displaying a 2-copy chromosomal integration of the transgenic construct at the site of the wildtype protease encoding gene, which was used as reference for subsequent long-read mapping.

A nested BLAST and filtering strategy was used, starting with a BLAST search of the raw long read datasets with either 2000 bp upstream or downstream of the chromosomal (wildtype) protease gene as queries. The hits were filtered as described above, and the filtered read dataset was subjected to a second round of BLAST, this time using pUB110 as query, after which hits were filtered again. The reads in the final read set were thus assumed to cover the putative site of chromosomal integration of the transgenic construct at the site of the wild-type protease gene. The read discovered during the analysis described in the previous paragraph was used as reference, to which all the other filtered reads were mapped with Minimap2 2.17 with the map-ont preset options [28]. Alignments were converted to BAM format, sorted and indexed with SAMtools 1.9, and visualized with Integrated Genomics Viewer 2.4.10 [29].

### 2.6. SNP Phylogeny and SNP Typing

To construct SNP-based phylogenies of the GMM isolates, all publicly available Illumina paired-end data of *B. velezensis* strains were collected via the ENA API (January 2021). The 42 retrieved datasets were combined with the Illumina paired-end data of the GM isolates, and a SNP matrix was obtained by running the CFSAN SNP pipeline [30] on processed unorphaned forward and reverse reads for each sample. The Unicycler hybrid assembly of isolate Pilsner1-2 was taken as reference since it contained the least fragmented assembly of all. The number of SNPs, without using any filters, was used to estimate the distance of the strains to the reference. Strains that showed more than 10,000 SNPs were excluded from further analysis, in accordance with the readme documentation of SnapperDB (github.com/phe-bioinformatics/snapperdb, accessed 15 August 2021). All GM strains of this study were retained and supplemented with two public datasets (Appendix A) using this criterion. The CFSAN pipeline was then rerun on the reduced sample set, with the same reference. The preserved (filtered) SNP matrix was used for evolutionary model selection through the web interface of SMS 1.8.4 [31]. Subsequently, a maximum-likelihood phylogenetic tree was constructed with PhyML 3.3.3:3.3.20170530+dfsg-2 [32] with the settings recommended by SMS: GTR substitution model, proportion of invariable sites fixed, 1 substitution rate category, and SPR tree topology search mode. One hundred bootstrap replicates were performed.

Additionally, SNP addresses of the reduced sample set were extracted with PHEnix 1.4.1 [33] and SnapperDB 1.0.6 [33] with the same reference genome, as described by Nouws et al. [34]. The 7-number SNP addresses provide an isolate level hierarchical clustering nomenclature, whereby isolates sharing an increasing amount of SNPs are increasingly more closely related. For example, for a group of isolates sharing only the first SNP address number, each isolate in the group is less than 250 SNPs away from at least one other isolate in the group. Likewise, the SNP thresholds are 100, 50, 25, 10, 5 and 0 for the second to last SNP address numbers, respectively.

Lastly, to obtain pairwise SNP distances for the GM isolates of this study, the CFSAN pipeline was run as described above, but solely including the 10 GM isolates of this study, with the same reference, and converted to a SNP distance matrix with snp-dists 0.7.0 (github.com/tseemann/snp-dists, accessed on 14 August 2021).

### 2.7. Whole Genome Alignment-Based Comparison

For each isolate, the representative assembly obtained as described in Section 2.4 was employed. To facilitate interpretation, these assemblies were first manually pruned to retain only the largest contigs, accounting for at least 91.6% of the total assembly size in all cases (Appendix A). This also implied that all contigs carrying the transgenic plasmid were removed from the assemblies. Multiple genome alignment was done with progressiveMauve 20150213 [35] with default settings, and included pruned annotated assemblies of the 10 GMM isolates of this study, *B. velezensis* 10075, *B. velezensis* CBMB205 (Genbank: NZ_CP011937), which is the representative strain for this species in the RefSeq database, and a number of arbitrarily chosen *B. velezensis* strains, for which complete assemblies were publicly available (see Appendix A for a complete list). The resulting alignments were visualized and investigated in Mauve viewer.

ProgressiveMauve 20150213 with the --seed-family option was also employed for multiple genome alignment of a putative plasmidic phage and a selection of known linear plasmidic phages, for which details are provided in Appendix A. Prior to the alignment, the MauveContigMover tool of Mauve 20150213 was employed to put all the sequences in the same orientation and facilitate their comparison.

## 3. Results and Discussion

### 3.1. Isolation of Viable GM B. velezensis Producing Protease from FE Products, Long- and Short-Read WGS

Previously, Fraiture et al. [2] described the genomic characterization of a viable *B. velezensis* GMM, isolated from a commercial FE product, more specifically a protease (i.e., ‘Pureferm’ in Table 1). Based on these findings, qPCR methods were developed that could subsequently be used to detect the presence of DNA from this unauthorized protease-producing GMM in other commercial FE products (Table 1) [6]. Among these FE products, in contrast to the Pureferm product labeled as protease, two were labeled as alpha-amylase and one was labeled as an enzyme mixture, among which were alpha-amylase and protease. From all these FE products, viable isolates of a protease-producing *B. velezensis* GMM could be obtained, which was reported at the EU level via the Rapid Alert System for Food and Feed (RASFF) notifications. Following confirmation by qPCR that they carried the transgenic protease construct, ten isolates from four different FE samples were selected for genomic characterization. An overview of the strains included in this study, with the FE products and batches from which they were isolated, is presented in Table 1.

A limitation of the former study of Fraiture et al. [2] was that the WGS data consisted of short-reads (Illumina) only, resulting in a relatively fragmented assembly. Although the presence of a GMM was firmly established, the generated short-read assembly did not allow to confidently determine the genomic location and copy number of the transgenic construct. Over the past few years, third-generation sequencing technologies, such as Oxford Nanopore Technologies (ONT), have become widely accessible, and increasingly performant, facilitating the generation of much longer reads, albeit at a lower accuracy, compared to reads from the dominant second-generation sequencing platform Illumina. Consequently, hybrid assembly methods have been developed that allow to exploit the strengths of both short- and long-read sequencing technologies, i.e., the low sequencing error rate of the Illumina technology to obtain very accurate assemblies, and the capacity of long reads to bridge and resolve repetitive regions in the genome [36]. A (more) complete, accurate genome assembly could allow to unambiguously establish the location of the transgenic construct in a GMM, i.e., whether it is plasmidic or genomic, and its copy number. A recent study by Berbers et al. [9] highlighted the added value of such a ‘hybrid’ approach for genomic characterization of GMM. Hence, we chose to sequence genomic DNA of isolates from the protease-producing GM *B. velezensis* using the Illumina and ONT technologies, in order to determine the location and copy number of the transgenic construct, and determine the relationship between the different FE products contaminated with this GMM.

The two MinION flow cells, each containing five samples, delivered 5,387,509 reads with a read length N50 of 7646 bp and median read quality of 12.1 and 7,211,191 reads with a read length N50 of 2603 bp and median read quality of 11.8, respectively. After demultiplexing, the number of reads per sample ranged from 101,521 to 2,422,322, and the read length N50 per sample varied between 1951 and 12,809 bp. Read quality was very similar for the different samples. Per-sample metrics are provided in Appendix A.

Raw short-read input varied from 461,014 to 729,454 per sample (Appendix A) and was generally of very high quality. In the GC distribution plot of the FastQC report (Appendix A) of all isolates, the main peak was located at 45–46%GC, corresponding to the expected value of 45.6–47%GC reported for complete chromosome assemblies of *B. velezensis* in the RefSeq database (date retrieved: 03/2021). Besides this main peak, a conspicuous broad secondary peak was present at 28–37%GC, in line with the expected value of 35.7% for plasmid pUB110 (Genbank: M19465.1). This pronounced secondary peak was congruent with the presence of pUB110 as a free high-copy number plasmid, and/or the integration of a substantial number of pUB110 copies into the chromosome.

### 3.2. Genetic Characterization of GMM

The overview of the genetic characterization of the protease-producing *B. velezensis* GMM provided in this section was obtained by integrating results from different assembly approaches, complemented with additional in-depth investigation of raw long read data. A more detailed description of each of these analyses and their contribution to the full characterization is available in the Appendix A.

#### 3.2.1. Characterization of GMM Host Strain

Figure 1 depicts graphs of the Unicycler, Canu, and hybridSPAdes assemblies of sample Pilsner1-2, which was chosen as representative isolate because it presented the least fragmented hybrid assembly. Assembly graphs for the other isolates were similar for each assembly method, and are shown in Appendix A. The main assembly stats for all assemblies are presented in Appendix A.

With respect to the chromosome, the publicly available *B. velezensis* genome that was most similar to that of the GMM isolates of this study based on web-based blastn, was that of *B. velezensis* 10075, a strain isolated from food lobster sauce in China. This strain was therefore also included in the comparative genomic analysis (see Section 3.3.2). Next to the chromosome, all but one of the protease GMM isolates carried an extrachromosomal natural element, which is most likely a plasmidic prophage. No AMR genes were detected on the chromosome or putative plasmidic prophage of the GMM isolates. In all the isolate assemblies, the only AMR genes present were associated with the transgenic construct (see Section 3.2.2).

#### 3.2.2. The Transgenic Modification Is Present as an Episomal High-Copy Plasmid

Based on the integration of the assembly results with the additional analyses, the hybrid assembly via Unicycler was able to most accurately reflect the ground truth of the protease GMM isolates, i.e., with the transgenic construct on a high-copy free plasmid of 6756 bp in length (Figure 1A) (Appendix A). Supporting evidence for this genetic make-up was provided by short-read sequencing and assembly of a plasmid extract, details of which are provided in the Appendix A, and by the presence of a distinct peak at 6.6–6.8 kbp visible in the read length histogram of raw pUB110-matching long-reads, corresponding to the length of the recombinant plasmid (Appendix A) (Appendix A). The high-copy nature of the plasmid in *Bacillus* spp. is described in the literature [37], and is supported by the high read-depth for the plasmid-covering reads, as compared to the read-depth observed for genome-covering reads for all the isolates (results not shown).

However, because Unicycler failed to run to completion for 7 out of 10 samples, an alternative hybrid assembly tool, hybridSPAdes, was also run on the samples. HybridSPAdes yielded very fragmented assemblies, showing integration of the transgenic construct into the chromosome, contrary to the Unicycler assemblies (Figure 1B) (Appendix A). Therefore, Canu, a long-read only assembly method, was also executed, which supported the presence of the transgenic construct as an extrachromosomal structure, but not of the integration that was apparent in the HybridSPAdes assemblies. Noteworthy, contrary to the single-copy circular plasmid revealed in the Unicycler assemblies, all Canu assemblies exhibited one or more contigs carrying concatemers, i.e., multiple contiguous head-to-tail copies, of the recombinant plasmid (Figure 1C,D) (Appendix A).

Figure 2 depicts the genetic structure of the high-copy recombinant plasmid, derived from the pUB110 shuttle plasmid. Part of the pUB110 sequence was deleted, and replaced by a recombinant insert (Appendix A) encoding a protease gene that is 100% identical to the natural protease gene in the chromosome of the *B. velezensis* host strain. The transgenic construct harbored two AMR genes: *aadD*, conferring kanamycin and neomycin resistance, and *ble*, conferring bleomycin resistance. All elements required for normal replication [37] had remained intact, indicating that the vector was designed to be episomal rather than integrative [1] (Appendix A).

#### 3.2.3. The Transgenic Plasmid Shows Sporadic, Unstable Integration into the Chromosome and Its Replication Is Disturbed, Leading to Accumulation of Linear Plasmid Concatemers

Our investigation revealed two additional findings of interest. First, we found evidence that despite the transgenic construct being primarily carried on a free high-copy plasmid, unstable transient chromosomal integration did appear to occur to some limited extent. This was supported by a limited number of raw long reads, observed for all isolates, that exhibited one to two contiguous copies of the plasmid integrated at the site of the natural chromosomal copy of the protease encoding gene (Figure 3). The fact that this vector can sporadically integrate into the host genome is an additional reason for concern, as this presumably homology-driven integration event may similarly occur in related naturally occurring strains that have taken up the vector by horizontal gene transfer.

Secondly, we observed for all isolates an abundance of raw long-reads displaying so-called plasmid concatemers, representing linear high molecular-weight head-to-tail copies of the recombinant pUB110-derived plasmid. As stated above, one or more contigs displaying these concatemers were present in all of the long-read (Canu) assemblies, while they did not show up in the hybrid (Unicycler) assemblies. The most likely explanation for this concatemer-related phenomenon is disturbed plasmid replication in the cells, since previous studies have reported that pUB110-replication can be disrupted under certain conditions, e.g., insertion of foreign DNA into pUB110, leading to accumulation of high-molecular-weight head-to-tail plasmid multimers in e.g., *B. subtilis* [39] (Appendix A).

Overall, our results highlight that correct assembly of a GMM, even with long-read data, can pose specific challenges, hence requiring a particularly careful approach including multiple additional ad hoc analyses, instead of relying on the output of one particular assembler.

### 3.3. Comparative Analysis and Source Tracing

To determine the genomic relationship between the isolates from different FE products, and assess whether they potentially originate from a common source, two approaches were used to perform an in-depth comparative analysis.

#### 3.3.1. SNP Phylogenetic Analysis and Typing Indicate the Isolates Share a Common Source

SNP phylogenetic analysis and SNP typing were combined to infer the phylogenomic context of the isolates, and quantify their differences. Based on an explorative screening analysis that included all available *B. velezensis* strains for which Illumina paired-end data was publicly available (Appendix A), only samples that differed by fewer than 10,000 SNPs from the reference isolate Pilsner1-2 were included in the SNP-based phylogeny and SNP typing, retaining only two public datasets in total. Although approximately 338 *B. velezensis* genome assemblies were publicly available in NCBI (date accessed March 2021), paired-end Illumina data were available for only 42 strains.

The resulting SNP phylogenetic tree and accompanying SNP addresses are shown in Figure 4 (CFSAN pipeline metrics in Appendix A). All 10 isolates clustered together monophyletically, and were at least 250 SNPs away from the two other strains in the tree. The GM isolates shared always at least the first four digits of their SNP addresses, i.e., they were always within at least 25 SNPs of one other isolate. Isolates originating from the same FE product clustered together, as expected. Moreover, Crystal and Pure isolates, which originate from the same supplier (see Table 1), also clustered together. This underpins the sensitivity of this method, and its potential for source-tracking of unauthorized GMM. Pilsner2-1 and Pilsner2-2 shared the same SNP address, i.e., no SNPs were found between both isolates. The same was true for Crystal-1 and Crystal-2. Pilsner1-1 and Pilsner1-2 shared the first six digits of their SNP address, implying that they differed by fewer than five SNPs. Inspection of their pairwise SNP distances (Appendix A) showed that they actually differed by only one SNP. For Pure-1 vs. Pure-2, and Cob9-1 vs. Cob9-2, which shared only the first four digits of their SNP address, the difference was more notable, i.e., 16 and 10 SNPs difference, respectively (Appendix A).

Recently, Pightling et al. [40] proposed a set of criteria to evaluate whether bacterial isolates share a common source, based on SNP phylogenies. The study focused on pathogenic bacteria and tracing of outbreak sources, but the general framework can equally well be applied to GMM, thus providing a useful and objective guideline. The criteria are threefold: (1) isolates cluster together monophyletically, which was the case here as the GM isolates were clearly delineated from the two background samples; (2) bootstrap support for the monophyletic cluster should be >0.89, and was 100% in this case; and (3) the isolates differ from each other by at most a few tens of SNPs, and in this case all GM isolates belonged to the same 25 SNP-cluster, with the largest pairwise distance between any two isolates being 21 SNPs (Appendix A). Since all three criteria were fulfilled, this strongly supports that the isolates derive from the same parental strain used to manufacture the different GM fermentation products.

#### 3.3.2. Whole Genome Comparison Supports Results from SNP-Based Analysis

A multiple genome alignment was performed to evaluate potential genomic rearrangements between isolates that could be missed by short read-based SNP analysis because of the short-read lengths. Overall, this analysis confirmed that the isolates displayed very high mutual sequence conservation, and that *B. velezensis* 10075 was highly related to the GMM isolates (Appendix A). This is illustrated by the presence of a putative prophage in the isolates, and in strain 10075, which was clearly absent from all other included *B. velezensis* strains (Appendix A). Although this region was practically identical in the 10 isolates, it showed marked sequence divergence compared to the homologous region in strain 10075, indicating a certain degree of evolutionary distance between 10075 and the GM isolates.

Inspection of the genomic region around the chromosomal location of the protease encoding gene in the isolates and in strain 10075, indicated that this region was highly conserved in all 10 isolates compared to 10075, and no signs of recombination or presence of elements/genes that are not present in 10075 were detected, supporting the absence of genetic modification of the host genome within this region (Appendix A).

At certain sites, indications of rearrangements were however found between the isolates. Phage characterization of the representative Unicycler assembly of sample Pilsner1-2 (Appendix A), and cross-comparison with the multiple genome alignment of the isolates (Appendix A) revealed that prophages appeared to be abundantly present, accounting for ~416 kbp in total, or ~9.5%, of the draft genome of isolate Pilsner1-2. All apparent intra-contig break points coincided with prophage sequences, and most, but not all, contig boundaries were also found to occur at prophage sites, thus the apparent rearrangements likely represented misassembly artefacts. Alternatively, these could represent authentic rearrangement events, due to very recent prophage proliferation activity, or a combination of both. Nevertheless, in one specific case, clear signs of a large-scale rearrangement were found, i.e., in the Crystal isolates (Appendix A), where exactly the same locally collinear block (LCB) of ~116 kbp, not associated with a prophage, was found to be rearranged compared to all other isolates and strain 10075.

## 4. Conclusions

In this study, the presence of viable GMM, carrying AMR genes, was observed in four commercial FE products, which was reported at the EU level via the Rapid Alert System for Food and Feed (RASFF) notifications (Table 1). By combining a variety of bioinformatics approaches using short-read and/or long-read data, backed up with experimental evidence, we were able to thoroughly characterize 10 GM bacterial isolates, obtained from the different commercial FE products.

We could confirm that the three GM isolates for which a Unicycler assembly could be obtained, carry a recombinant plasmid, derived from pUB110, with an insertion containing the wild-type protease encoding gene. The presence of a peak at 6.6–6.8 kbp in the read length histograms of all the isolates strongly supports that this plasmid is present in the other GM isolates as well. Furthermore, the recombinant plasmid is partly present in the form of linear plasmid multimers, probably due to a disturbance of the normal plasmid replication mechanism, and it is likely only transiently integrated into the chromosome. This study highlights the added value of a hybrid approach via WGS, since neither short-read nor long-read sequencing methods solely were able to correctly resolve the relatively complex genomic constitution of the GM isolates.

These findings raise serious food safety and public health concerns, which are emphasized by the fact that the AMR genes are harbored on a free plasmid, increasing the risk for spreading of AMR via horizontal gene transfer. They also highlight the need for thorough genetic characterization of GMM since the location, i.e., chromosomal or plasmidic, of the transgenic construct and associated AMR genes, is important information to consider for safety evaluation.

Furthermore, we could conclude that the GMM isolates from different FE products are genetically almost identical, and that they most probably originate from the same parental GMM strain. In particular, SNP-phylogenetic analysis based on short-read mapping was able to resolve the phylogenetic relationship between the isolates. To our knowledge, this study is the first to demonstrate the potential of such an approach for source-tracking of GMM. The outcome of the phylogenetic analysis was confirmed by whole-genome alignment-based comparison, while additionally demonstrating misassemblies or rearrangements that were caused by prophage sequences, and also the existence of one large-scale rearrangement in the isolates obtained from one FE, which would have been left unnoticed otherwise.

## Figures and Tables

**Figure 1 foods-10-02637-f001:**
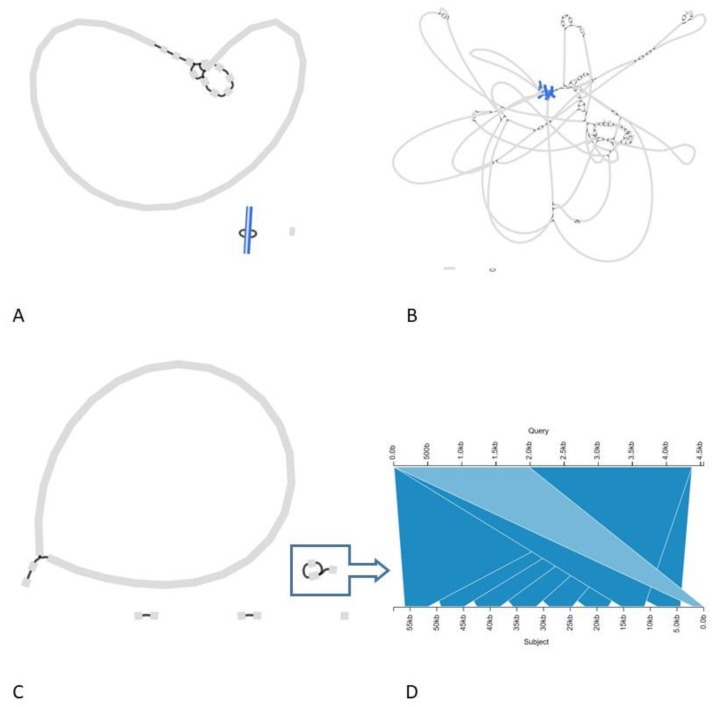
Assembly graphs of Unicycler, hybridSPAdes, and Canu assemblies of GM isolate Pilsner1-2. (**A**): Unicycler assembly of Pilsner1-2, with the pUB110 sequence on the plasmid contig in the assembly highlighted in blue, and the contig representing the putative plasmidic prophage highlighted in green. (**B**): HybridSPAdes assembly, with the pUB110 sequence inside the chromosome scaffold highlighted in blue, and the contig representing the putative plasmidic prophage highlighted in green. (**C**): Canu assembly with the three contigs that display recombinant plasmid concatemers framed in blue. (**D**): Kablammo visualization of the largest of the three contigs carrying concatemers of the Canu assembly. The top ruler represents the plasmid pUB110, and the bottom ruler represents the contig, containing 8 complete copies of the recombinant plasmid.

**Figure 2 foods-10-02637-f002:**
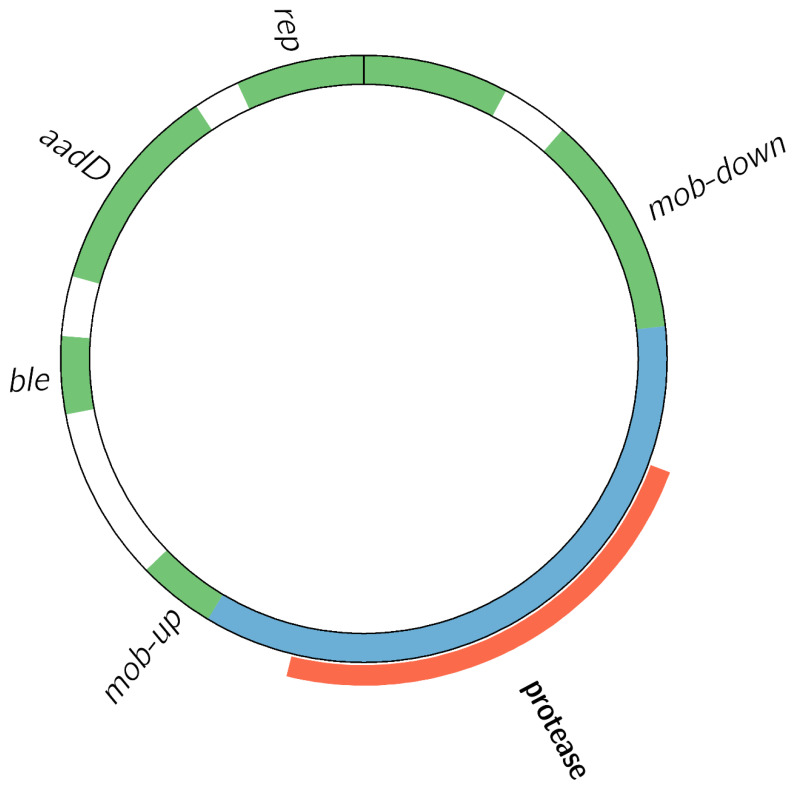
Map of the recombinant 6756 bp plasmid carried by the GM isolates. *aadD*: kanamycin and neomycin resistance gene, *ble*: bleomycin resistance gene, *rep*: replicase, *mob*: mobilization protein. A part of the *mob* gene is absent compared to the original pUB110 vector, and is replaced with a recombinant insert of 2385 bp in length, highlighted in blue, encompassing a complete protease encoding gene, highlighted in orange, which is also present in the chromosome of the isolates, as well as in the chromosome of *B. velezensis* 10075. Figure created with Circos 0.69-6 [38].

**Figure 3 foods-10-02637-f003:**
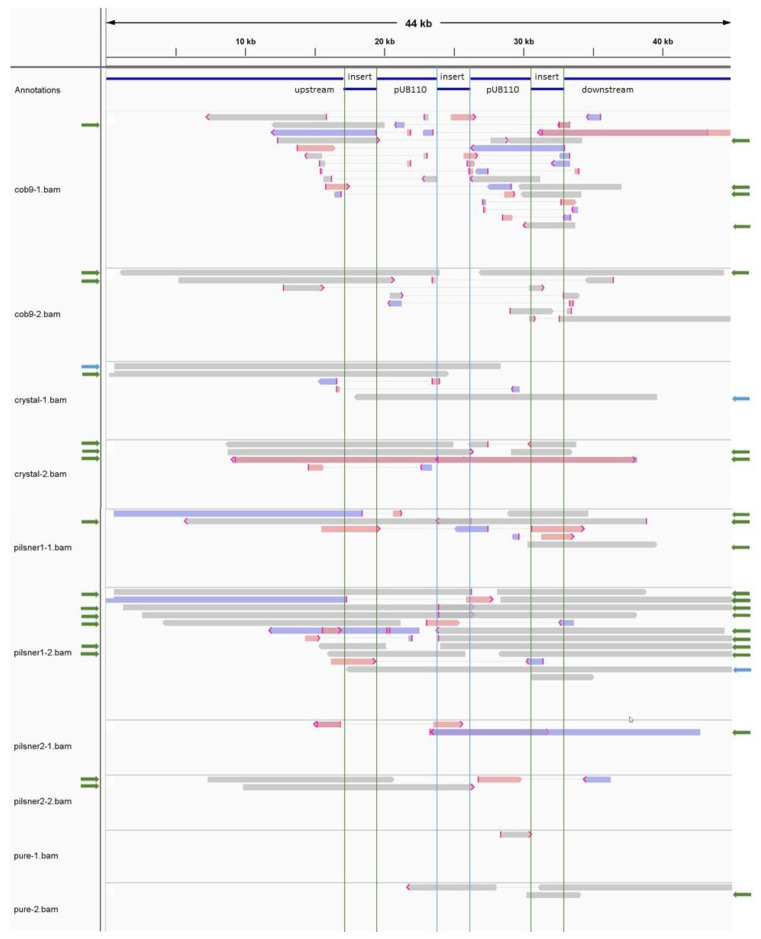
Alignment to reference read of ~44 kb, originating from isolate Crystal-1, displaying a two-copy integration of the recombinant plasmid. The alignment shows all raw long reads of all isolates (as labeled in the first column ‘Annotations’) that match pUB110 as well as chromosomal sequence upstream or downstream of the sequence used as insert in the transgenic construct, which carries the wild-type protease (hereafter referred to as ‘insert’). These reads are thus expected to cover the junction of chromosomal integration of the recombinant plasmid. If a supplementary alignment was reported, its connection to the primary alignment is shown via a thin grey line. The primary, or representative, alignment is the highest scoring alignment for a particular read. If the read does not align in its entirety, parts of the read sequence may be clipped, i.e., removed from the alignment. In some cases, (mostly) non-overlapping parts of the read may align at different positions to the reference. In that case, the non-primary alignment(s) is/are referred to as supplementary alignment(s). Primary and supplementary alignments that are inverted compared to each other are marked in red (forward alignment compared to the reference) and blue (reverse complement alignment). Clipped regions are hidden, and, if they are >30 bp, flagged with a red mark. For clarity, secondary alignments (same part of the read from the primary alignment, but aligning at another location to the reference), mismatched bases (variants) and short indels are hidden. Green vertical lines mark the edges of the first and third copy of the insert, which is present three times in the reference, as a consequence of the double plasmid integration event that is represented by the reference. The blue vertical lines mark the end of the first, and beginning of the second, integrated pUB110 copy in the reference read, with in between another (third) copy of the insert. If a continuous alignment extends beyond two adjacent green lines, it supports a chromosomal integration of at least one pUB110 copy. These reads are marked with a green arrow. If the alignment extends beyond both blue lines, in addition to two adjacent green lines, it supports the presence of (at least) two contiguous integrated pUB110 copies. These reads are flagged with a blue arrow. Figure created with IGV.

**Figure 4 foods-10-02637-f004:**
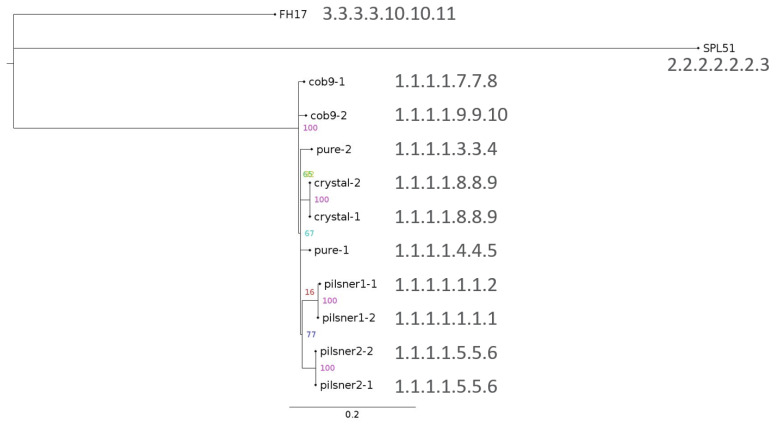
SNP-based phylogenetic tree, visualized with FigTree (github.com/rambaut/figtree, 29 October 2021), with associated SNP addresses. Both for SNP phylogeny and SNP typing, the Unicycler assembly of isolate Pilsner1-2 was used as reference. The scale is expressed as average substitutions per site in the SNP matrix. Node values represent bootstrap support values.

**Table 1 foods-10-02637-t001:** Overview of GMM isolates, and their associated commercial FE products and RASFF identifiers included in this study. Batch numbers denote different commercial FE product packages.

Commercial FE Product (Supplier)	Associated RASFF	Labeled Enzymes	Application	Evaluated Batch	Obtained GMM Isolates
Alpha-amylase enzyme 4 g (Coobra)	RASFF2020.2577	Alpha-amylase	Distillery	1	Cob9-1
					Cob9-2
Crystalmash (The Alchemist’s Pantry)	RASFF2019.3332	Alpha-amylase, Protease, Cellulase, Xylanase, Beta-glucanase	Distillery, Brewing, Grain processing	1	Crystal-1
					Crystal-2
Enzyme 4 g (Pilsner)	RASFF2020.2582	Alpha-amylase	Distillery, Brewing	1	Pilsner1-1
					Pilsner1-2
				2	Pilsner2-1
					Pilsner2-2
Pureferm (The Alchemist’s Pantry)	RASFF2019.3332	Neutral protease	Beer and other cereal based beverages; Bakery products and other cereal based products	1	Pure-1
					Pure-2

## Data Availability

Raw data and assemblies were deposited in the European Nucleotide Archive under study accession number PRJEB44065.

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
