# Peer review of "Characterization of Genetically Modified Microorganisms Using Short- and Long-Read Whole-Genome Sequencing Reveals Contaminations of Related Origin in Multiple Commercial Food Enzyme Products"

_foods, 2021, doi:10.3390/foods10112637_

Round 1

Reviewer 1 Report

The authors recall the repeated detection of GMMs in microbial fermentation products, among which several related to B. velenzis. They characterized B. velenzis from several food enzyme products using both long and short reads. Authors show that strains share a plasmid with resistance genes and have a common origin: a GM strain. They claim the interest of using hybrid WGS and SNP for GMM characterization and source-tracking.

In addition to the interest for GM concern, this paper highlights the possibility of plasmid concatemers, therefore the increased risk of horizontal gene transfer of this kind of GM component.

Major concerns:

Minor concerns:

Line 108, Table 1: Authors do not address the reason why they studied several strains in a batch. Considering their results, was it usefull to mention Cob9-2, Crystal-2, Pilsner1-2, Pilsner2-2 and Pure-2? Considering close results of “paired samples”, cannot you reduce your analysis to 5 GM strains instead of 10? Cannot observed differences be explained only by analysis heuristics?

line 184: “The resulting hits were filtered from the raw data with the filterbyname.sh script of bbtools 38.34 (source-forge.net/projects/bbmap), with default settings.”

please add “except ‘include=true’” because the default setting of the script is supposed to filter out the reads with matching names, it is not supposed to keep them (according to documentation).

line 217: missing SnapperDB citation provided line 227

line 251: “ProgressiveMauve 20150213 with default settings was also employed for multiple genome alignment of a putative plasmidic phage and a selection of known linear plasmidic phages”

Phage alignement may be improved by using –seed-family option of progressiveMauve. Results will probably be very similar, but slightly better.

Line 253: extra space between S and 6 in “Figure S 6”

line 297: “ranged from 101,521-2,422,322”, use ‘en dashes’, not hyphen for ranges. Found on many lines.

line 298: “varied between 1,951-12,809 bp”, same remark

line 363: “The transgenic construct harbored two AMR genes: aadD, conferring kanamycin and neomycin resistance, and ble, conferring bleomycin resistance.”

Wasn’t it highlighed by Authors earlier that no AMR gene was found in assemblies? Line 335?: “No AMR genes were detected on the chromosome or putative plasmidic prophage of the GMM isolates.” To clarify description I think Authors must add line 335 that “AMR genes are found on high copy plasmids, as described line…”. Otherwise it is a little bit confusing.

Line 409: “in all the long-read only Canu assemblies”, please reformulate to make it easier to understand at the first reading.

In supplementary material, figure S2, line 324: please improve the figure to avoid or clarify crossed arrows

In supplementary material, Figure S6: line 346: remove extra space between S and 6 in “Figure S 6”

line 354 to 357: it is a useful clarification but Authors can:
- Reorder contigs using MauveReorder (either in interface or in command line MauveCM in conda env), then
- re-align with progressiveMauve (and possibly –seed-family for sensitivity) to improve alignment of reordered contigs. This way, Authors can display all assemblies in the same orientation, facilitating comparisons by user.

In supplementary material, Figure S7: line 359: remove extra space between S and 7 in “Figure S 7”

In supplementary material, Figure S8: line 362: remove extra space between S and 8 in “Figure S 8”

In supplementary material, Figure S9: line 368: remove extra space between S and 9 in “Figure S 9”

In supplementary material, Figure S10: line 380: remove extra space between S and 10 in “Figure S 10”

In supplementary material, Figure S11: line 388: remove extra space between S and 11 in “Figure S 11”

line 392: replace “(black rectangle)” by “(black vertical rectangle)”

In supplementary material, Figure S12:line 395: remove extra space between S and 12 in “Figure S 12”

In supplementary material, Figure S13: line 405: remove extra space between S and 13 in “Figure S 13”

In supplementary material, Figure S14: line 416: remove extra space between S and 14 in “Figure S 14”

line 436: extra ’/’ I suppose in “2’/’ signifies that the plasmid “, otherwise authors must clarify the meaning of this legend.

In supplementary material, Table S2: line 440: remove extra space between S and 2 inTable S 2”

In supplementary material, Table S3: line 452: remove extra space between S and 3 inTable S 3

In header of the table, first “no reads” must be in uppercase for first letter.

In supplementary material, Table S4: line 464: remove extra space between S and 4 inTable S 4

In supplementary material, Table S5: line 472: remove extra space between S and 5 inTable S 5

In supplementary material, Table S6: line 493: remove extra space between S and 6 inTable S 6

In supplementary material, Table S7: line 506: remove extra space between S and 7 inTable S 7

In supplementary material, Table S8: line 519: remove extra space between S and 8 inTable S 8

Author Response

We want to thank the reviewer for thoroughly reading the manuscript, and for the useful comments and suggestions. Please see the attachment for our responses to the comments.

Reviewer 2 Report

Manuscript Foods 2021, 10, x. https://doi.org/10.3390/xxxxx

Characterization of genetically modified microorganisms using short- and long-read whole-genome sequencing reveals contaminations of related origin in multiple commercial food enzyme products

This is a well-constructed study and written manuscript. The characterization of Gmm for detecting contamination in food and other products. The research design appears appropriate. The results are laid out carefully and convey the right amount of information. The main concern is the supplemental material. It is quite dense and extensive. It is larger than the manuscript is itself. Some of it could be eliminated and some incorporated into the manuscript. The editor will need to make a judgment on this suggestion.

Author Response

We agree that the Supplementary information (SI) is quite extensive, especially compared to the main manuscript. We have chosen deliberately to keep the main manuscript concise to focus on the case study, and to move details, mainly concerning the bioinformatics analysis, to the SI. The SI is mainly of interest to a specialized audience (i.e. in bioinformatics and microbial genomics), but would still be required to reproduce the analysis we have conducted (explaining why it is quite extensive). We believe that the proposed organization of the main manuscript and SI suits the scope of Foods, by ensuring that the principal results of our study (the unauthorized presence of a viable GMM in different food enzymes most likely originating from a common source) are accessible to a very broad audience of scientists active in food sciences, many of whom have less expertise with genomics and might get lost in the advanced bioinformatics analyses we performed. We would therefore propose to keep this organization. However, if the editor concludes that we should shift or eliminate content of the SI, we will comply with this suggestion and move parts of the SI to the main manuscript.